# An instrument for measuring job satisfaction (VIJS): A validation study for community pharmacists in the context of the COVID-19 pandemic in Vietnam

**Thuy Thi Phuong Nguyen, Giang Thi Huong Truong, Huong Thi Thanh Nguyen, Cuc Thi Thu Nguyen, Dai Xuan Dinh**[ID]*, **Binh Thanh Nguyen**

Faculty of Pharmaceutical Management and Economics, Hanoi University of Pharmacy, Hanoi City, Vietnam

* daidinh.hup@gmail.com

**Data Availability Statement:** All relevant data are within the paper and its Supporting Information files.

## Abstract

### Background

Job satisfaction is one of the main factors creating and bringing about work motivation, productivity, and efficiency as well as decreasing job-hopping and job turnover. No previous studies have been conducted to assess job satisfaction for community pharmacists in Vietnam.

### Objective

This research was conducted to develop and validate an instrument used to measure community pharmacists' job satisfaction in the context of the COVID-19 pandemic in Vietnam.

### Methods and results

A total of 351 pharmacists participated in this survey. Data were analyzed using R software version 4.2.0. The final instrument (VIJS) has 34 items divided into six factors which were determined via a parallel analysis (including physical working conditions, work nature, income and other benefits, management policies and managers, relationships with coworkers and customers, and learning and advancement opportunities). VIJS's internal consistency was excellent (Cronbach's alpha = 0.97, Omega total = 0.98, split-half reliability = 0.985, and composite reliability>0.8). Two-week test-retest reliability results (intraclass correlation coefficient for the overall instrument: 0.97, for six factors: 0.865–0.938) demonstrated the consistency of the VIJS when the same test was repeated on the same sample (62 pharmacists) at different points in time. The Confirmatory Factor Analysis was employed to assess the construct validity. The VIJS was a good fit to a six-factor model (Chisq/df = 2.352, Comparative Fit Index = 0.937, Tucker-Lewis Index = 0.929, Standardized Root Mean Square Residual = 0.042, and Root Mean Square Error of Approximation = 0.062). VIJS's good convergent and discriminant validity was demonstrated via Average Variance Extrated>0.5 and the Heterotrait-Monotrait ratio of correlations<0.85.

**Funding:** The author(s) received no specific funding for this work.

**Competing interests:** The authors have declared that no competing interests exist.

## Conclusions

The VIJS possesses good reliability and validity and can be used to measure community pharmacists' job satisfaction.

## Introduction

In this modern world, a plethora of occupations have arisen and therefore people have numerous opportunities to select the career they like to work. Job satisfaction is one of the main factors affecting job performance and employee turnover. A high level of job satisfaction can help to create and bring about work motivation, productivity, and efficiency as well as decrease job-hopping and job turnover. For professional jobs such as health workers and community pharmacists, job satisfaction also affects patient care [1, 2].

The magnitude of job satisfaction of pharmacists around the world varies considerably. The satisfaction rate of Indian pharmacists was only 17.5% [3]. In Baghdad (Iraq), community pharmacists were extremely dissatisfied with their colleagues (86.0%), recognition (72.2%), and public respect (42.0%) [4]. In Arizona (the United States), 25% of chain community pharmacists were dissatisfied with the recognition they got for good work [5]. A moderate level of job satisfaction was witnessed among Malaysian community pharmacists. Among 13 factors used to measure job satisfaction, perceived societal status, professional responsibility, job security, and work demand were the job characteristics that pharmacists were least satisfied with [6].

Many facets contributing to the job satisfaction of pharmacists were reported, including stress, workload, supervisors/managers, coworkers, advancement opportunities, and income, to mention but a few [2, 7–9]. The most common reasons for pharmacists' job quitting and leaving intentions were dissatisfaction with their working environment, low remuneration, and a lack of promotion opportunities [10]. In comparison with pharmacists working in hospitals and primary care sectors, community pharmacists were less satisfied with their work [11, 12]. Remuneration ranked first in the aspects of pharmacists' work that they felt least satisfied with, regardless of their age, sex, or the type of sector [12]. For Swedish pharmacists, a lower level of job satisfaction was associated with increasing years in the current job position [13]. Increasing workloads and losing income can be repercussions of the COVID-19 pandemic and affect the job satisfaction of not only pharmacists but also a myriad of other occupations [12, 14].

In Vietnam, up to the year 2021, only one study was carried out to measure job satisfaction for clinical pharmacists in hospitals [15]. No previous studies have been conducted to assess job satisfaction for community pharmacists in Vietnam. There are approximately 82,000 community pharmacists who are working in medicine outlets all over Vietnam. The outbreak of the COVID-19 pandemic has given rise to myriad negative impacts on people's lives (especially for health workers and community pharmacists). This research was conducted to develop and validate a self-administered instrument to measure the job satisfaction of community pharmacists in the context of the COVID-19 pandemic in Vietnam.

## Methods

### Setting and participants

This cross-sectional study was conducted in Hanoi, Vietnam after approved by the Institutional Review Board and Ethical Committee of Hanoi University of Pharmacy (reference

number 125/QĐ-DHN, 22-06/PCT-HĐĐĐ). The study population was community pharmacists working in medicine outlets in the Hanoi capital (roughly 12.000 pharmacists). The inclusion criteria for selecting participants included community pharmacists licensed to sell medicines (medicine sellers), being Vietnamese natives, at least 18 years old, and working in medicine outlets opened on the days of data collection. By reason of the COVID-19 outbreak, pharmacists were recruited using a non-probability convenience sampling technique.

For studies using Exploratory Factor Analysis (EFA) and Confirmatory Factor Analysis (CFA), as per the rules of thumb, a suggested sample size for one item/question is from 3 to 20 people. A sample of 300 people is adequate for factor analysis [16–21]. Among 450 pharmacists approached, 351 pharmacists (response rate: 78.0%) voluntarily answered a questionnaire used to measure their job satisfaction and informed consent was obtained from all of them. Two weeks after the first survey, 62 invited pharmacists agreed to answer this questionnaire one more time. The minimum sample size for evaluating the test-retest reliability was 51 people (with an expected intraclass correlation coefficient (ICC): 0.75, a type 1 error rate: 5%, a 95% confidence interval width: 0.2) [22].

## Instrument development and data collection

A pool of items was developed by reviewing numerous articles and scales/instruments involving job satisfaction [3–6, 12, 23–26]. After this initial review, seven main factors which can affect the job satisfaction of community pharmacists were selected (including physical working conditions, work nature, income and other benefits, management policies and managers, relationships with coworkers, relationships with patients/customers, and learning and advancement opportunities). The number of items for these factors was 6, 11, 6, 12, 6, 5, and 7 items, respectively. The initial instrument included 53 items involving job satisfaction.

A Vietnamese self-administered questionnaire was designed (including a short introduction about this survey, a confirmation of voluntary participation in this research, and the two following main parts). The first part included eight questions involving the background information of pharmacists and their medicine outlets (including sex, year of birth, marital status, level of education, working experience, time of working per day, salary per month, and the average number of customers per day of outlets). The second part was the initial instrument including 53 items with 5-point Likert scale responses ((1) Strongly disagree, (2) Disagree, (3) Normal, (4) Agree, and (5) Strongly agree). A higher score represented a higher level of job satisfaction. This questionnaire was designed on Google Forms. From March to April 2022, data collectors paid a visit to medicine outlets and invited pharmacists to directly answer this questionnaire on an Ipad. After the process of data analysis, nineteen items were removed because their ICCs were low (<0.5) (7 items), item loadings (EFA) were low (<0.5) (8 items), and/or Cronbach's alphas ($\alpha$) for factors increased if these items were dropped (6 items). The final instrument (VIJS) includes 34 items (Table 1).

## Data analysis

After collected, data were extracted into a Microsoft Excel file and analyzed using R software version 4.2.0 (packages *tidyverse*, *psych*, *irr*, *EFAtools*, *GPArotation*, *performance*, *lavaan*, *semTools*, and *semPlot*). The mean (standard deviation) and number (percentage) were used to report information for continuous variables and categorical variables, respectively.

Regarding the reliability of the VIJS, internal consistency was assessed through $\alpha$, split-half reliability, composite reliability (CR), and McDonald's Omega (Omega Total $\omega_t$ and Omega Hierarchical $\omega_h$) with the higher values indicating the better consistency. $\alpha \geq 0.7$ and $\geq 0.8$ indicate acceptable and good internal consistency, respectively [20, 27]. CR>0.70 is interpreted a

**Table 1. Item names, item-total correlation, and two-week test-retest reliability of the VIJS.**

| | Items | Item-total correlation | Intraclass correlation coefficient (ICC) |
|---|---|---|---|
| A1 | I am satisfied with the facilities of my medicine outlet. | 0.78 | 0.68 (0.47–0.807) |
| A2 | My workplace is spacious, clean, and airy. | 0.82 | 0.682 (0.472–0.808) |
| A3 | I am working in safe conditions. | 0.82 | 0.624 (0.373–0.774) |
| A4 | My medicine outlet has full equipment for professional work (such as a computer with an Internet connection, an air conditioner...) | 0.81 | 0.523 (0.214–0.712) |
| A5 | There are enough means to help prevent the spread of disease/pandemic in my medicine outlet (such as face masks, hand sanitizers...) | 0.84 | 0.548 (0.252–0.727) |
| B1 | I am satisfied with my current job position. | 0.84 | 0.758 (0.597–0.854) |
| B2 | The work that I am doing is suitable for the qualifications and skills that I was trained. | 0.90 | 0.778 (0.631–0.866) |
| B3 | I can use and promote my capacity well at work. | 0.89 | 0.658 (0.429–0.794) |
| B4 | Professional work (such as selling medicines, and patient counseling) fulfills my career aspirations. | 0.87 | 0.564 (0.277–0.738) |
| B5 | I can meet and interact with many people. | 0.78 | 0.608 (0.35–0.764) |
| C1 | I am satisfied with my current income. | 0.90 | 0.766 (0.614–0.859) |
| C2 | My salary is commensurate with my current job position. | 0.95 | 0.677 (0.466–0.805) |
| C3 | The bonus and remuneration I received are commensurate with my work performance. | 0.94 | 0.608 (0.348–0.764) |
| C4 | Additional income (bonus) is distributed equally. | 0.88 | 0.589 (0.318–0.752) |
| D1 | The managers always care about the lives of employees. | 0.87 | 0.783 (0.64–0.87) |
| D2 | The managers treat all employees in my medicine outlet fairly and suitably. | 0.87 | 0.655 (0.426–0.793) |
| D3 | The managers have enough capacity and knowledge to effectively manage and monitor the work at the medicine outlet. | 0.81 | 0.809 (0.683–0.885) |
| D4 | The managers trust me in my work. | 0.85 | 0.614 (0.358–0.767) |
| D5 | The managers always support and guide all employees enthusiastically. | 0.90 | 0.575 (0.294–0.744) |
| D6 | The managers always listen to opinions and quickly respond to employees. | 0.88 | 0.65 (0.42–0.789) |
| D7 | The managers always praise/encourage employees when they do a good job. | 0.85 | 0.531 (0.219–0.719) |
| E1 | My coworkers are competent and knowledgeable. | 0.80 | 0.591 (0.325–0.752) |
| E2 | My coworkers are hospitable and friendly people. | 0.81 | 0.67 (0.451–0.802) |
| E3 | My coworkers and I usually share experiences and help each other in our work. | 0.89 | 0.591 (0.319–0.754) |
| E4 | My coworkers are trustworthy. | 0.90 | 0.72 (0.536–0.831) |
| E5 | My coworkers and I care for and help each other in life. | 0.86 | 0.807 (0.679–0.884) |
| E6 | I am satisfied with my relationship with patients/customers. | 0.84 | 0.532 (0.219–0.719) |
| E7 | Patients/customers have a respectful attitude towards me. | 0.74 | 0.518 (0.197–0.711) |
| F1 | I am satisfied with the training and advancement opportunities at my outlet. | 0.86 | 0.731 (0.553–0.838) |
| F2 | I have been trained in professional knowledge and skills regularly. | 0.86 | 0.729 (0.55–0.836) |
| F3 | I have many opportunities to participate in professional training courses (such as clinical pharmacy, drug information, drug consulting and using...) | 0.85 | 0.682 (0.471–0.809) |
| F4 | I will have many opportunities to advance to a higher position if I work hard. | 0.86 | 0.727 (0.535–0.838) |
| F5 | I have opportunities to study and promote my capacity at work. | 0.84 | 0.808 (0.681–0.884) |
| F6 | My medicine outlet has fair and clear promotion policies. | 0.83 | 0.76 (0.601–0.856) |

ICC: p-values for all items < 0.001, except for items A4 (p = 0.00196), A5 (p = 0.00108), D7 (p = 0.00197), E6 (p = 0.00196), and E7 (p = 0.00271).

good reliability [20, 28]. To evaluate split-half reliability, items were randomly divided into two parts and the correlation between these two parts was computed. $\omega_h$ and $\omega_t$ which should be over 0.70 can be used to present the proportion of the variance of the total scores due to the general factor and due to all factors, respectively [28–30]. ICC was used to evaluate the test-retest reliability—the reliability of a test measured at different points in time (<0.5: poor, 0.50–0.75: moderate, and >0.75: good reliability) [31].

The construct validity of the VIJS was assessed using EFA and CFA (with principal axis factoring analysis, promax rotation). CFA, a type of structural equation modeling (SEM), can be used to test how well the measured items represent a set of theoretical latent factors. For studies with more than 250 people and instruments having 30 items or more, the main fit indices used for evaluating the CFA/SEM model include the p-value expected for the Chi-squared test (<0.001), Comparative Fit Index (CFI>0.92), Tucker-Lewis Index (TLI>0.92), Standardized Root Mean Square Residual (SRMR<0.08), and Root Mean Square Error of Approximation (RMSEA<0.07) [20]. Other fit indices which can be used include the Goodness-of-Fit statistic and the Adjusted Goodness-of-Fit statistic (GFI, AGFI>0.9: good, 0.8–09: acceptable); Root Mean Square Residual (RMR<0.08); Incremental Fit Indices (IFI>0.9); the Parsimony Goodness-of-Fit Index and the Parsimonious Normed Fit Index (PGFI, PNFI>0.5) [32].

The external validity of the VIJS was evaluated via convergent and discriminant validity. The Average Variance Extracted (AVE) which is used to measure convergent validity should be at least 0.50. The Heterotrait-Monotrait ratio of correlations (HTMT—the ratio of the between-trait correlations to the within-trait correlations) should be lower than 0.85 [20].

## Results

### Participants' characteristics

A total of 351 community pharmacists participated in this survey. More than four-fifths of participants were women and about two-fifths of them were married. Most pharmacists were aged from 21 to 35 years old (84.33%). Their average age was 28.85±7.24 years old. About two-fifths of the participants were university graduates. Pharmacists' working experience in medicine outlets was mostly less than 10 years (82.05%). In medicine outlets, pharmacists mainly worked eight hours a day and earned a monthly salary of four million Vietnam dongs or more. On average, the number of customers of medicine outlets ranged from 30 to 100 people per day (71.79%) (Table 2).

### Construct validity (EFA and CFA)

The final VIJS included 34 items which were divided into six factors: Factor A: *Physical working conditions*: 5 items, Factor B: *Work nature*: 5 items, Factor C: *Income and other benefits*: 4 items, Factor D: *Management policies and managers*: 7 items, Factor E: *Relationships with coworkers and customers* (7 items), and Factor F: *Learning and advancement opportunities*: 6 items (Table 1). To assess the factorability of the data, Bartlett's test of sphericity was statistically significant (with Chisq = 10,876.8, p-value<0.001) and the Kaiser-Meyer-Olkin measure of sampling adequacy (KMO) = 0.954>0.5. The number of factors to extract (n = 6 factors) was determined using a parallel analysis (Fig 1). Item loadings (cutoff = 0.5) from a principal axis factoring analysis with promax rotation were presented in Table 3. The communality values of all 34 items were higher than 0.46. For six extracted factors, the cumulative percentage of variance accounted for was 69.25>50.0.

Results from CFA showed that the final six-factor model had a good fit with CMIN/df (Chisq/df) = 2.352 (<3), p-value = 0.000, CFI = 0.937 (>0.92), TLI = 0.929 (>0.92), GFI = 0.841 (>0.8), AGFI = 0.812 (>0.8), NNFI = 0.929 (>0.9), NFI = 0.896 (>0.8), IFI = 0.937 (>0.9), RNI = 0.937 (>0.9), RMR = 0.023 (<0.05), SRMR = 0.042 (<0.05), RMSEA = 0.062 (<0.07), (95%CI: 0.057–0.067, p-value = 0.000), PNFI = 0.801 (>0.5), and PGFI = 0.710 (>0.5). In addition, six factors had moderate/strong correlations with one another (correlation coefficients: from 0.56 to 0.84). Each factor provided an acceptable explanation of variation found among its items with standardized loadings from 0.66 to 0.94 and error variances from 0.12 to 0.57 (Fig 2). Overall, the data indicated that the VIJS has good construct validity.

**Table 2. Participants' characteristics (N = 351 pharmacists).**

| Participants' characteristics | | Number (n) | Percentage (%) |
|---|---|---|---|
| Sex | Male | 48 | 13.68 |
| | Female | 303 | 86.32 |
| Age (years old) | 21–25 | 159 | 45.30 |
| | 26–30 | 95 | 27.07 |
| | 31–35 | 42 | 11.97 |
| | 36–40 | 30 | 8.55 |
| | 41 and above | 25 | 7.12 |
| Marital status | Married | 150 | 42.74 |
| | Not married | 201 | 57.26 |
| Level of education | University and higher | 148 | 42.17 |
| | College and lower | 203 | 57.83 |
| Working experience (years) | < 5 | 213 | 60.68 |
| | 5–9 | 75 | 21.37 |
| | 10–14 | 33 | 9.40 |
| | ≥ 15 | 30 | 8.55 |
| Time of working per day (hours) | < 8 | 29 | 8.26 |
| | 8 | 223 | 63.53 |
| | > 8 | 99 | 28.21 |
| Salary per month (million Vietnam dongs)* | < 2 | 5 | 1.42 |
| | 2 –less than 4 | 42 | 11.97 |
| | 4–6 | 129 | 36.75 |
| | > 6 | 175 | 49.86 |
| The average number of customers per day (people) | < 30 | 66 | 18.80 |
| | 30 –less than 50 | 151 | 43.02 |
| | 50–100 | 101 | 28.77 |
| | > 100 | 33 | 9.40 |

*: Exchange rate: one million Vietnam dongs = 43.78763 US$ (for the first day of data collection)

## The reliability

The good internal consistency of the VIJS was demonstrated through the high values of split-half reliability (0.985), α (0.97), $\omega_t$ (0.98), and $\omega_h$ (0.83) for the overall instrument with 34 items. The Spearman-Brown split-half reliability coefficients and α for all six factors were higher than 0.85. CRs for all six factors were also higher than 0.8. The item-total correlation coefficients of all 34 items ranged from 0.74 to 0.95. Furthermore, the VIJS possessed a high level of two-week test-retest reliability with an ICC of 0.974 for the overall 34-item instrument and from 0.896 to 0.938 for six factors (p-values<0.001) (Tables 1 and 4).

## The external validity

For all six factors, AVEs were higher than 0.50. The HTMTs of all pairs of factors were lower than 0.85 (Table 4). As a result, the convergent and discriminant validity of the VIJS was good.

## Discussion

Job satisfaction is an important factor that positively affects people's working motivation, commitment, performance, and productivity; negatively links to complaints, tardiness/absenteeism, job-hopping, and job turnover. In Vietnam, there are no previous studies conducted to

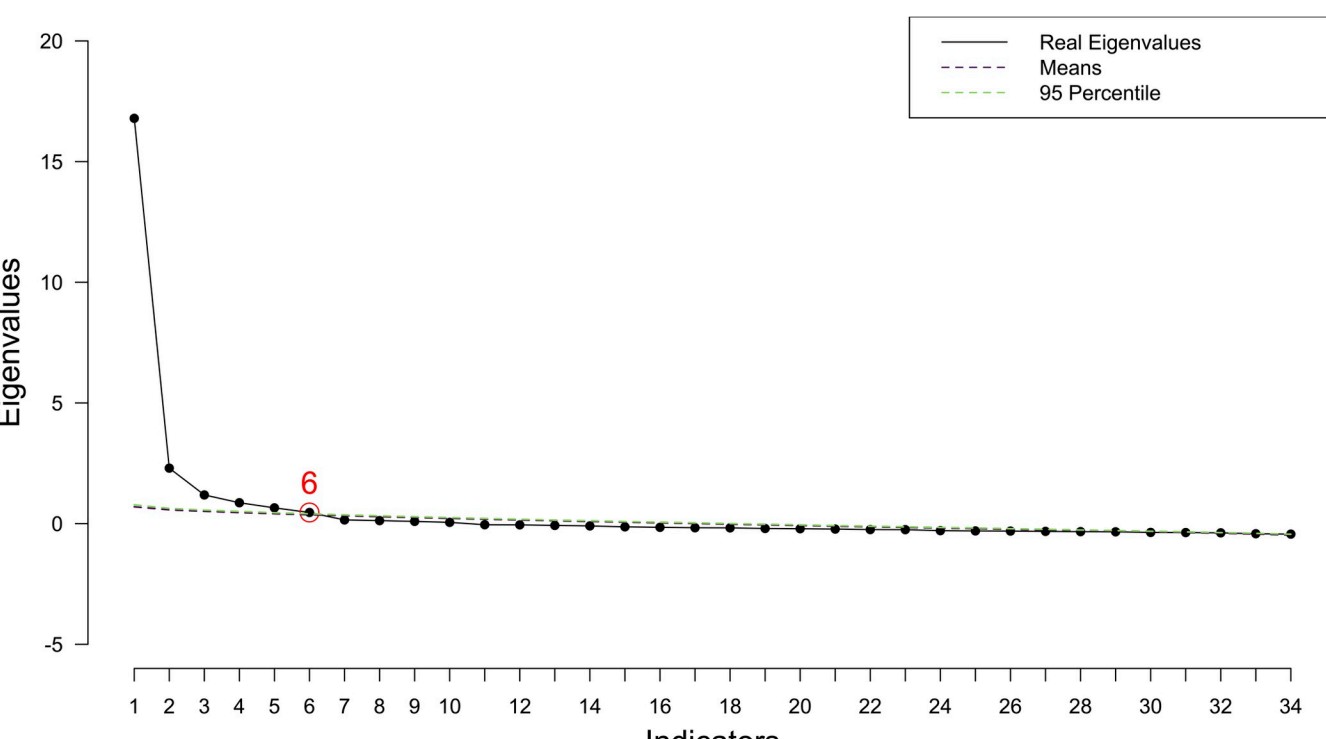

**Fig 1. Determining the number of factors to extract using parallel analysis.** EFA: Exploratory Factor Analysis.

assess job satisfaction for Vietnamese community pharmacists. In this research, the authors strived to develop and validate a instrument (VIJS) that can be used to measure job satisfaction for community pharmacists. Results from this survey with the participation of 351 pharmacists demonstrated that this instrument possesses good reliability and validity.

The VIJS possesses good internal consistency and test-retest reliability. α for six factors (from 4 to 7 items per factor) ranged from 0.87 to 0.94, consistent with results from several studies using the Warr-Cook-Wall 10-item scale: α = 0.90 (for pharmacists in the United Kingdom) [25] and α = 0.87 (for community pharmacists in Baghdad, Iraq) [4]; JS-Q (34 items, 8 domains, α from 0.751 to 0.930) [33] but higher than the result of a study for pharmacists in Malaysia (15-item scale, α = 0.78) [34]. Although α is the most widely used method for evaluating internal consistency, the omega coefficient is a better choice than α [35]. High values of omega coefficients ($\omega_t$ and $\omega_h \geq 0.8$), the split-half reliability coefficients (>0.86), and CRs>0.8 were additional evidence for the good reliability of the VIJS. The item-total correlation coefficient of an item (ranging from 0 to 1) indicates whether the response of this item is consistent with its factor. The item-total correlation coefficients of all 34 items (from 0.74 to 0.95) showed that VIJS's items measuring different factors delivered consistent scores of job satisfaction. In addition, the results of a two-week test-retest reliability (ICC for the overall instrument: 0.97, ICCs for each factor: from 0.865 to 0.938, ICCs for each item: >0.5) showed the consistency of this instrument when the same test was repeated on the same sample (62 people) at different points in time (two weeks after the first survey).

In CFA, the Chi-squared statistic is one of the absolute fit indices used to measure how well the model reproduces the observed data and the p-value is commonly higher than 0.05. However, numerous factors can impact this test, including the number of observations (participants)

**Table 3. Promax-rotated item loadings of 34 items in the final six-factor model.**

| Items | Promax-rotated item loadings (cutoff = 0.5) | | | | | | Communality |
|---|---|---|---|---|---|---|---|
| | Factor A | Factor B | Factor C | Factor D | Factor E | Factor F | |
| A1 | 0.617 | | | | | | 0.520 |
| A2 | 0.666 | | | | | | 0.607 |
| A3 | 0.623 | | | | | | 0.618 |
| A4 | 0.673 | | | | | | 0.594 |
| A5 | 0.821 | | | | | | 0.706 |
| B1 | | 0.579 | | | | | 0.688 |
| B2 | | 0.793 | | | | | 0.771 |
| B3 | | 0.860 | | | | | 0.766 |
| B4 | | 0.709 | | | | | 0.676 |
| B5 | | 0.558 | | | | | 0.556 |
| C1 | | | 0.745 | | | | 0.743 |
| C2 | | | 0.902 | | | | 0.881 |
| C3 | | | 0.930 | | | | 0.878 |
| C4 | | | 0.696 | | | | 0.696 |
| D1 | | | | 0.697 | | | 0.704 |
| D2 | | | | 0.796 | | | 0.758 |
| D3 | | | | 0.626 | | | 0.608 |
| D4 | | | | 0.669 | | | 0.696 |
| D5 | | | | 0.890 | | | 0.793 |
| D6 | | | | 0.743 | | | 0.749 |
| D7 | | | | 0.763 | | | 0.723 |
| E1 | | | | | 0.633 | | 0.591 |
| E2 | | | | | 0.662 | | 0.635 |
| E3 | | | | | 0.846 | | 0.800 |
| E4 | | | | | 0.909 | | 0.804 |
| E5 | | | | | 0.899 | | 0.735 |
| E6 | | | | | 0.767 | | 0.668 |
| E7 | | | | | 0.574 | | 0.464 |
| F1 | | | | | | 0.596 | 0.727 |
| F2 | | | | | | 0.642 | 0.690 |
| F3 | | | | | | 0.711 | 0.666 |
| F4 | | | | | | 0.878 | 0.737 |
| F5 | | | | | | 0.703 | 0.672 |
| F6 | | | | | | 0.703 | 0.623 |

Factor A: *Physical working conditions*, Factor B: *Work nature*, Factor C: *Income and other benefits*, Factor D: *Management policies and managers*, Factor E: *Relationships with coworkers and customers*, Factor F: *Learning and advancement opportunities*.

and the number of items of instruments/scales. In this study, the number of participants was 351 (>250) and the number of items of the VIJS was 34 (>30). The Chi-squared test had a p-value<0.001 (significant p-value), consistent with information from a book from Hair [20]. Other absolute fit indices (GFI>0.8, RMSEA<0.08, RMR<0.05, SRMR<0.05), incremental fit indices (NFI>0.89, NNFI>0.9, TLI>0.92, CFI>0.92, RNI>0.9), and parsimony fit indices (AGFI>0.8, PNFI>0.5, PGFI>0.5) revealed that the VIJS was a good fit to a six-factor model.

Besides good construct validity, the good convergent and discriminant validity of the VIJS can be demonstrated via AVEs>0.5 and HTMTs<0.85. AVE is the mean variance extracted for

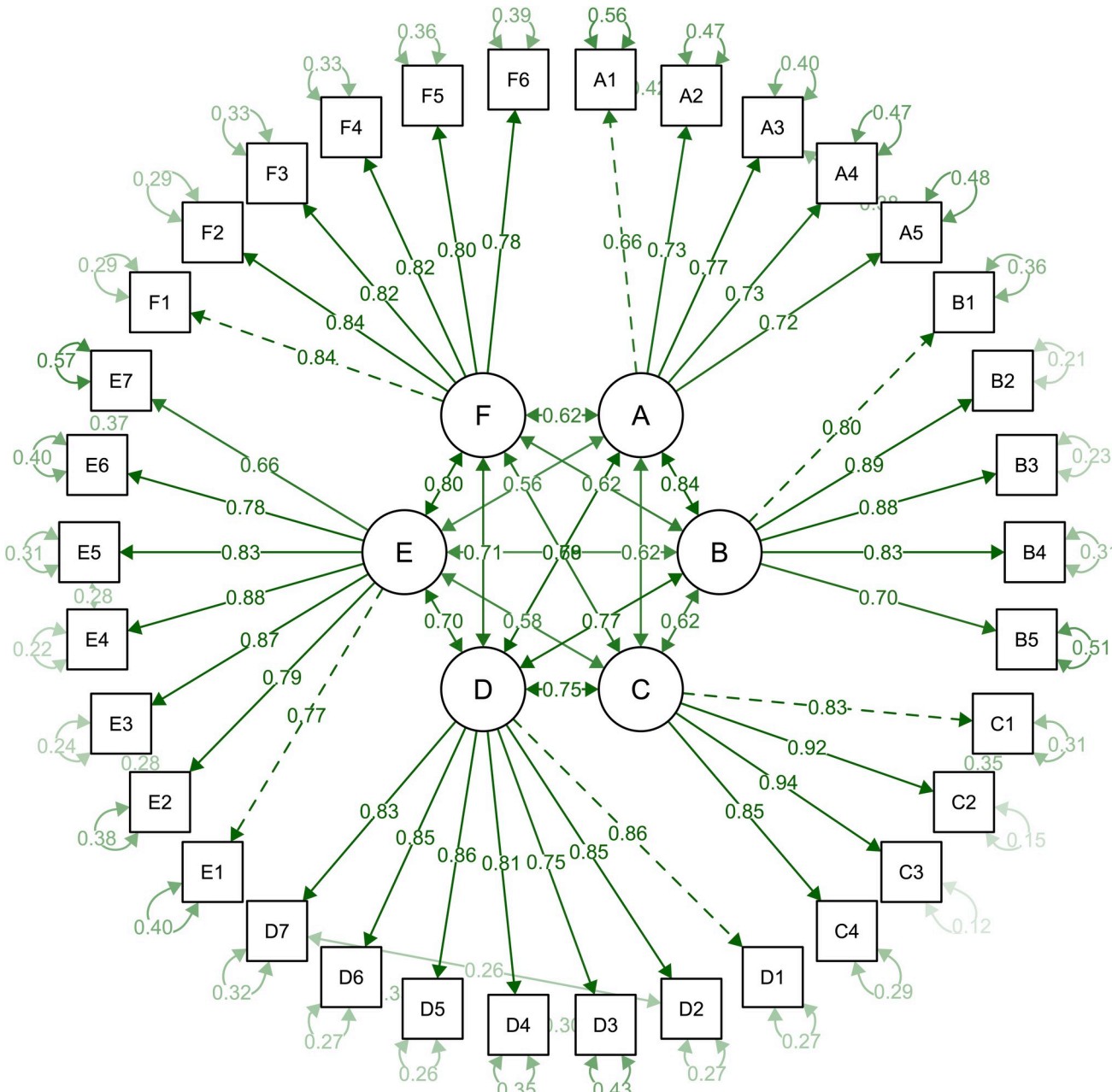

**Fig 2. Confirmatory factor analysis for the six-factor model of the VIJS.** Factor A: *Physical working conditions*, Factor B: *Work nature*, Factor C: *Income and other benefits*, Factor D: *Management policies and managers*, Factor E: *Relationships with coworkers and customers*, Factor F: *Learning and advancement opportunities.*

items loading on a construct/factor. AVE>0.5 represents that a latent variable/factor can explain more than 50% of the variance of its observed variables/items [20]. To assess the discriminant validity, the HTMT was used in this study. Numerous previous studies used the Fornell-Lacker criterion and the examination of cross-loadings. However, a simulation study revealed that in common study situations, the aforementioned criterion and testing did not reliably detect the lack of discriminant validity and authors recommended the use of a new criterion (HTMT) to assess the discriminant validity in variance-based structural equation modeling [36].

**Table 4. Main indicators used to assess the reliability and validity of the VIJS.**

| Factor | α | ω_t | ω_h | Split-half reliability | ICC | CR | AVE | HTMT | | | | | |
|---|---|---|---|---|---|---|---|---|---|---|---|---|---|
| | | | | | | | | A | B | C | D | E | F |
| A | 0.87 | 0.92 | 0.80 | 0.87 | 0.90 | 0.85 | 0.52 | 1.00 | | | | | |
| B | 0.91 | 0.92 | 0.89 | 0.92 | 0.91 | 0.91 | 0.68 | 0.82 | 1.00 | | | | |
| C | 0.94 | 0.95 | 0.91 | 0.95 | 0.92 | 0.94 | 0.79 | 0.60 | 0.66 | 1.00 | | | |
| D | 0.94 | 0.96 | 0.89 | 0.96 | 0.94 | 0.94 | 0.69 | 0.76 | 0.78 | 0,76 | 1.00 | | |
| E | 0.93 | 0.95 | 0.89 | 0.95 | 0.93 | 0.93 | 0.64 | 0.54 | 0.59 | 0.61 | 0.70 | 1.00 | |
| F | 0.92 | 0.95 | 0.87 | 0.94 | 0.94 | 0.92 | 0.67 | 0.59 | 0.62 | 0.69 | 0.71 | 0.79 | 1.00 |

α: Cronbach's alpha, $\omega_h$: McDonald's Omega Hierarchical, $\omega_t$: McDonald's Omega Total, ICC: Intraclass correlation coefficient, CR: Composite Reliability, AVE: Average Variance Extracted, HTMT: Heterotrait-Monotrait ratio of correlations

Factor A: *Physical working conditions*, Factor B: *Work nature*, Factor C: *Income and other benefits*, Factor D: *Management policies and managers*, Factor E: *Relationships with coworkers and customers*, Factor F: *Learning and advancement opportunities*.

ICC: p-values for all factors < 0.001

Overall, the final instrument includes 34 items divided into six factors: (1) *Physical working conditions*: 5 items, (2) *Work nature*: 5 items, (3) *Income and other benefits*: 4 items, (4) *Management policies and managers*: 7 items, (5) *Relationships with coworkers and customers*: 7 items, and (6) *Learning and advancement opportunities*: 6 items. Results from CFA and test-retest reliability demonstrated the good reliability and validity of the VIJS. However, this research has several limitations. Firstly, participants were selected using a convenience sampling strategy in the Hanoi capital. As a consequence, the study sample cannot be representative of Vietnamese community pharmacists. Furthermore, because of the length of the initial instrument, the authors did not use other job satisfaction scales/instruments to measure job satisfaction and evaluate the VIJS's concurrent validity.

## Conclusions

Results from this study demonstrated that the VIJS (34 items, six factors) is an instrument possessing good reliability and validity. Future studies can use this instrument to measure the job satisfaction of community pharmacists in other locations or countries.

## Supporting information

**S1 File. The questionnaire.**
(DOCX)

**S2 File. Data file.**
(XLSX)

## Acknowledgments

The authors would like to thank Quynh Diem Dieu and Anh Minh Hoang, two students from the Hanoi University of Pharmacy, who assisted us in data collection. We also appreciate all 351 community pharmacists who enthusiastically participated in this research.

## Author Contributions

**Conceptualization:** Thuy Thi Phuong Nguyen, Giang Thi Huong Truong.

**Data curation:** Thuy Thi Phuong Nguyen, Giang Thi Huong Truong, Dai Xuan Dinh.

**Formal analysis:** Thuy Thi Phuong Nguyen, Cuc Thi Thu Nguyen, Dai Xuan Dinh.

**Investigation:** Thuy Thi Phuong Nguyen, Giang Thi Huong Truong, Cuc Thi Thu Nguyen, Dai Xuan Dinh.

**Methodology:** Thuy Thi Phuong Nguyen, Giang Thi Huong Truong, Huong Thi Thanh Nguyen, Cuc Thi Thu Nguyen, Dai Xuan Dinh, Binh Thanh Nguyen.

**Project administration:** Thuy Thi Phuong Nguyen, Huong Thi Thanh Nguyen, Dai Xuan Dinh, Binh Thanh Nguyen.

**Resources:** Thuy Thi Phuong Nguyen.

**Software:** Dai Xuan Dinh.

**Supervision:** Thuy Thi Phuong Nguyen, Huong Thi Thanh Nguyen, Dai Xuan Dinh, Binh Thanh Nguyen.

**Validation:** Thuy Thi Phuong Nguyen, Dai Xuan Dinh.

**Visualization:** Dai Xuan Dinh.

**Writing – original draft:** Dai Xuan Dinh.

**Writing – review & editing:** Thuy Thi Phuong Nguyen, Giang Thi Huong Truong, Huong Thi Thanh Nguyen, Cuc Thi Thu Nguyen, Dai Xuan Dinh, Binh Thanh Nguyen.

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
