## [Decision Letter · Decision Letter 0]

6 Jul 2022

PONE-D-22-15763An instrument for measuring job satisfaction (VIJS): a validation study for community pharmacists in the context of the COVID-19 pandemic in VietnamPLOS ONE

Dear Dai Xuan Dinh

Thank you for submitting your manuscript to PLOS ONE. After careful consideration, we feel that it has merit but does not fully meet PLOS ONE’s publication criteria as it currently stands. Therefore, we invite you to submit a revised version of the manuscript that addresses the points raised during the review process.

We look forward to receiving your revised manuscript.

Kind regards,

Jahanpour Alipour, Ph.D.

Academic Editor

PLOS ONE

Journal Requirements:

Additional Editor Comments:

Dear author

-Please check and correct the entire manuscript carefully in terms of grammar and typography.

-For better formatting of the manuscript, the following link contains guidelines for writing the information on the title page and the body of the manuscript. It is suggested to see these guidelines to set different headings in different sections of the manuscript.

Reviewers' comments:

Reviewer's Responses to Questions

**Comments to the Author**

1. Is the manuscript technically sound, and do the data support the conclusions?

Reviewer #1: Yes

Reviewer #2: Yes

2. Has the statistical analysis been performed appropriately and rigorously? 

Reviewer #1: Yes

Reviewer #2: Yes

3. Have the authors made all data underlying the findings in their manuscript fully available?

Reviewer #1: Yes

Reviewer #2: Yes

4. Is the manuscript presented in an intelligible fashion and written in standard English?

Reviewer #1: Yes

Reviewer #2: No

5. Review Comments to the Author

Reviewer #1: The manuscript technically sound, and the data support the conclusions. The statistical analysis has been performed appropriately. The conclusions drawn appropriately based on the data presented. The manuscript presented in clear and comprehensible fashion and written in standard English.

Reviewer #2: This research is very well conducted and i recommend for publication however i would like to make few suggestions..

There re few typographic mistakes which need to be corrected all over the manuscript.

VIJS need to be explained more to make it easy for readers and why you select this instrument.

The factors used in different tables also need more explanation.

Figure 2 need explanation.

6. PLOS authors have the option to publish the peer review history of their article (what does this mean?). If published, this will include your full peer review and any attached files.

Reviewer #1: **Yes: **Thaer Abdelghani

Reviewer #2: No

---

## [Author Response · Author response to Decision Letter 0]

16 Aug 2022

REBUTTAL LETTER

Dear editors and reviewers.

First and foremost, we want to express our deep gratitude to the editors and reviewers who gave us valuable and practical advice. We carefully read your comments and revised our manuscript. Our responses are written in blue color. 

One change in the revised manuscript which I should mention here is our affiliations. In July 2022, our university was re-structured and the names of departments/faculties were changed. The name of our department was also changed from “Department of Pharmaceutical Management and PharmacoEconomics” into “Faculty of Pharmaceutical Management and Economics”.

Thank you so much for your assistance. 

Best regards. 

Dai Xuan Dinh

Hanoi University of Pharmacy

......................................................................................................

Journal Requirements

We strived to follow these guidelines.

We checked the reference list, revised some references, and deleted one reference: “Barnett CW, Kimberlin CL. Development and validation of an instrument to measure pharmacists' satisfaction with their jobs and careers. Am J Pharm Educ. 1986 Spring;50(1):5-14. PMID: 10281557.”

If there are any references that are inappropriate, please let us know.

3. Please check and correct the entire manuscript carefully in terms of grammar and typography.

For better formatting of the manuscript, the following link contains guidelines for writing the information on the title page and the body of the manuscript. It is suggested to see these guidelines to set different headings in different sections of the manuscript.

We checked all over our manuscript and revised grammar/typographic mistakes.

Reviewers' comments

Reviewer #1: 

The manuscript technically sound, and the data support the conclusions. The statistical analysis has been performed appropriately. The conclusions drawn appropriately based on the data presented. The manuscript presented in clear and comprehensible fashion and written in standard English.

Thank you for your compliments. We did our utmost to conduct this research.

Reviewer #2: 

This research is very well conducted and i recommend for publication however i would like to make few suggestions.

1. There are few typographic mistakes which need to be corrected all over the manuscript.

We checked all over our manuscript and revised typographic mistakes.

2. VIJS need to be explained more to make it easy for readers and why you select this instrument.

We planned to conduct a study to investigate the job satisfaction of Vietnamese community pharmacists. However, no instruments with Vietnamese versions could be found. As a result, we developed and validated this instrument in phase 1. And then, we will use this instrument to interview community pharmacists all over Vietnam (phase 2).

3. The factors used in different tables also need more explanation.

The factors used in different tables are the same. The VIJS includes the six following factors:

Factor A: Physical working conditions, 

Factor B: Work nature, 

Factor C: Income and other benefits, 

Factor D: Management policies and managers, 

Factor E: Relationships with coworkers and customers, 

Factor F: Learning and advancement opportunities.

These factors were named after we used EFA with promax rotation. Items for each factor can be seen in Table 1. 

4. Figure 2 need explanation.

The explanation of Figure 2 can be seen in lines 189-192. In addition, some relevant information mentioned in above paragraphs (such as information in lines 168-172) was not repeated.

---

## [Decision Letter · Decision Letter 1]

17 Oct 2022

An instrument for measuring job satisfaction (VIJS): a validation study for community pharmacists in the context of the COVID-19 pandemic in Vietnam

PONE-D-22-15763R1

Dear Dr. Dai Xuan Dinh,

We’re pleased to inform you that your manuscript has been judged scientifically suitable for publication and will be formally accepted for publication once it meets all outstanding technical requirements.

Kind regards,

Jahanpour Alipour, Ph.D.

Academic Editor

PLOS ONE

Additional Editor Comments (optional):

Reviewers' comments:

Reviewer's Responses to Questions

**Comments to the Author**

1. If the authors have adequately addressed your comments raised in a previous round of review and you feel that this manuscript is now acceptable for publication, you may indicate that here to bypass the “Comments to the Author” section, enter your conflict of interest statement in the “Confidential to Editor” section, and submit your "Accept" recommendation.

Reviewer #1: All comments have been addressed

Reviewer #2: All comments have been addressed

2. Is the manuscript technically sound, and do the data support the conclusions?

Reviewer #1: Yes

Reviewer #2: Yes

3. Has the statistical analysis been performed appropriately and rigorously? 

Reviewer #1: Yes

Reviewer #2: Yes

4. Have the authors made all data underlying the findings in their manuscript fully available?

Reviewer #1: Yes

Reviewer #2: Yes

5. Is the manuscript presented in an intelligible fashion and written in standard English?

Reviewer #1: Yes

Reviewer #2: Yes

6. Review Comments to the Author

Reviewer #1: The manuscript technically sound, and the data support the conclusions. The manuscript presented in clear and comprehensible fashion and written in standard English.

Reviewer #2: All the correction suggested was addressed properly and the necessary changes are made in the manuscript. I am satisfied with the changes and recommend for publication.

7. PLOS authors have the option to publish the peer review history of their article (what does this mean?). If published, this will include your full peer review and any attached files.

Reviewer #1: **Yes: **Dr. Thaer Abdelghani

Reviewer #2: No

---

## [Editor Report · Acceptance letter]

27 Oct 2022

PONE-D-22-15763R1 

An instrument for measuring job satisfaction (VIJS): a validation study for community pharmacists in the context of the COVID-19 pandemic in Vietnam 

Dear Dr. Dinh:

I'm pleased to inform you that your manuscript has been deemed suitable for publication in PLOS ONE. Congratulations! Your manuscript is now with our production department. 

Kind regards, 

on behalf of

Dr., Jahanpour Alipour 

Academic Editor

PLOS ONE